# Health, Environment or Taste? Using the Theory of Planned Behaviour to Predict Plant-Based Milk Consumption

**DOI:** 10.3390/foods14111970

**Published:** 2025-06-01

**Authors:** Indita Dorina, Ava Nikpour, Barbara Mullan, Hannah Uren

**Affiliations:** Behavioural Science and Health Research Group, enAble Research Institute, School of Population Health, Faculty of Health Sciences, Curtin University, Perth, WA 6102, Australia; indita.dorina@curtin.edu.au (I.D.); ava.nikpour@student.curtin.edu.au (A.N.); hannah.uren@curtin.edu.au (H.U.)

**Keywords:** plant-based milk, dairy alternatives, environmental behaviour, health behaviour, theory of planned behaviour

## Abstract

Dairy farming contributes significantly to global greenhouse gas emissions, highlighting the need for a dietary shift toward more sustainable products. Plant-based milks have gained popularity as a lower-calorie, more environmentally sustainable alternative to dairy milk. The aim of this study was to apply an adapted theory of planned behaviour (attitude, subjective norms and behavioural beliefs), expanded to include environmental, health and taste motives, to predict individuals’ intention to consume and consumption of plant-based milks. The participants (*N* = 286) completed a two-part online questionnaire assessing theory constructs at time one and behaviour at time two. Multiple regression analyses revealed that taste-focused attitude and health-focused behavioural beliefs significantly predicted intention to consume plant-based milks (*R*^2^ = 0.53, *p* < 0.001). Intention was the only significant predictor of plant-based milk consumption (*R*^2^ = 0.60, *p* < 0.001). These findings offer valuable insights into the motivators of plant-based milk consumption. Intentions should be targeted in interventions to encourage plant-based milk consumption by emphasising the taste value and by instilling individuals’ confidence to attain health benefits.

## 1. Introduction

Dairy products have a significant environmental footprint, with meat and milk being larger contributors to environmental burden than any other food group [1]. Emissions from animal greenhouse gases, feed production, land use and energy-intensive global supply chains contribute to nearly 60% of food-related greenhouse gas emissions and almost 20% of total global emissions [2]. Many people acknowledge that reducing dairy consumption would be beneficial for the environment and that plant-based milks offer a lower-calorie alternative to dairy products [3]. However, the dairy industry is only continuing to grow [4]. Therefore, it is important to understand the drivers behind sustainable food choices such as plant-based milk to promote environmentally friendly diets.

When choosing between plant-based and dairy milk, individuals often take into account the environmental impact, health benefits and taste of products [5]. Vegetarian and vegan diets, which often incorporate plant-based milk alternatives, have the largest potential to reduce global greenhouse gas emissions and lessen the environmental impact of the agriculture industry [6,7]. Plant-based milk has a significantly lower environmental impact compared to dairy milk as cultivating crops requires less water and land and generates approximately 3 times less greenhouse gas emissions than raising cattle [8]. However, while generally having a lower environmental impact than dairy milk, plant-based milks still present environmental concerns. The production of almond milk requires high water usage and is generally less sustainable than that of oat and soy milk [9]. Therefore, individuals may hold contrasting opinions on the environmental benefits of plant-based milks, especially due to negative media coverage of certain crops being associated with adverse effects on fragile ecosystems [9,10].

Alternatively, individuals may consume plant-based milk due to its health benefits, including its lower hormone and saturated fat contents than dairy milk, which are linked to cardiovascular disease and weight gain [3,7,11]. The widespread lactose intolerance within the global adult population also contributes to the increased popularity of plant-based milks [12]. But there are also consumer concerns regarding the nutritional quality of plant-based milks as they do not naturally contain the same high levels of calcium and protein as dairy milk, and its substitution may lead to undernutrition if these nutrients are not sourced elsewhere in one’s diet [13]. Plant-based milk also often contains added sugars, flavours or ingredients to improve its taste and texture, which affects the overall health profile [9]. Varying perceptions of the health benefits of plant-based milk can influence consumers’ decisions to choose these alternatives over dairy milk [5].

Ultimately, the literature suggests that taste remains the largest and most consistent factor of food choices, whether plant- or animal-based [14]. Individuals most often rationalise dairy consumption over plant-based alternatives due to preferences for taste [15]. Similarly, those who consume plant-based milk note taste as an important factor of their preference [16], and other attributes cannot compensate for disagreeableness in taste [17]. However, individuals have differing taste preferences for various plant-based milks [18]. Given the complexity of motives behind plant-based milk consumption, applying psychological theories of behaviour may help identify the key drivers of plant-based milk consumption to support sustainable dietary shifts.

### 1.1. Theory of Planned Behaviour

A widely used psychological model for understanding and predicting health and environmental behaviour is the theory of planned behaviour [19]. It was proposed in the theory that individuals’ intention to engage in a behaviour is the largest predictor of their engagement in the behaviour. Behavioural intentions are influenced by attitudes, subjective norms and perceived behavioural control. Attitudes refer to an individual’s evaluation of whether performing a behaviour can lead to positive or negative outcomes, subjective norms involve perceived social expectations to engage in a behaviour, and perceived behavioural control reflects the individual’s perception of the ease or difficulty of performing the behaviour. Perceived behavioural control is also proposed to directly influence engagement in a behaviour.

Several studies have demonstrated the effectiveness of the theory of planned behaviour in predicting intentions and behaviours related to food choice and dietary behaviours. For example, the theory was applied to predict exclusive breastfeeding [20], fruit and vegetable consumption [21] and sustainable diets [22,23]. More specifically to plant-based dietary behaviours, the theory was successfully applied to explore individuals’ willingness to adopt a plant-based diet [24] and intentions to consume plant-based milk yogurt substitutes [25] and soymilk [26]. Meta-analytic evidence suggests that positive attitudes, subjective norms and perceived behavioural control predict stronger intentions and engagement in certain diets [27]. However, perceived behavioural control may be more important for behaviours with near-future outcomes, rather than distant-future outcomes [28]. Behavioural beliefs, or beliefs about the consequences of performing a behaviour, may instead be more important for behaviours with distant-future outcomes [29]. As the health and environmental impacts of plant-based milk consumption may only be seen in the distant future, we suggest that behavioural beliefs rather than perceived behavioural control may instead be more influential.

Although current theory-based behaviour change interventions have shown some success in facilitating temporary change, evidence for their effectiveness in maintaining change over time has been limited [30]. Therefore, researchers have sought personalised intervention methods for their potential to be more effective in encouraging long-term behaviour change compared to interventions only considering generic motivations [31]. This is because individuals are more likely to internalise behaviours that align with their personal values [32]. Due to the diverse reasons individuals choose to consume plant-based milk, it may be useful to explore how the theory of planned behaviour variables, framed specifically to the three key drivers of plant-based milk consumption (environmental, health and taste motives), contribute to their decision making. Previous research has either applied the theory to examine the drivers of plant-based diets (e.g., [24,25]) or applied exploratory methods to identify the specific motives (e.g., [33,34]). However, no research to date has integrated these motives within a theoretical framework to offer a comprehensive, theory-driven investigation of plant-based milk consumption. This study addresses a gap in the research to extend the application of the theory of planned behaviour to consider the motives underlying plant-based milk consumption, which have not been explored previously. Therefore, more effective personalised behaviour change interventions that consider individuals’ personal motives to consume plant-based milks may be developed.

### 1.2. The Current Study

The aim of this study was to explore the factors influencing the intention to consume and consumption of plant-based milk using an adapted theory of planned behaviour: attitudes, subjective norms and behavioural beliefs. The second aim was to identify whether environmental-, health- or taste-focused attitudes; subjective norms; or behavioural beliefs were most likely to influence the intention to consume and consumption of plant-based milk see Figure 1). We hypothesised the following:

**H1.** 
*Attitude (H1a), subjective norms (H1b) and behavioural beliefs (H1c) will explain significant variance in intention to consume plant-based milk.*


**H2.** 
*Intention (H2a) and behavioural beliefs (H2b) will explain significant variance in plant-based milk consumption.*


**H3.** *Environmental-focused attitudes (H3a), subjective norms (H3b) and behavioural beliefs (H3c) will be more likely predict intention to consume plant-based milk than taste- and health-focused attitudes, subjective norms and behavioural beliefs, given the frequent association of plant-based milk with environmental sustainability* [3,35].

## 2. Materials and Methods

### 2.1. Procedure

Participants were recruited from August to November 2024 using convenience sampling via Prolific (an online recruitment platform) and an undergraduate participant pool. The undergraduate pool consisted of first- to third-year students taking psychology subjects. However, Prolific offers access to a diverse participant base and allows researchers to target individuals based on specific demographics or interests [36]. For this study, Prolific recruitment was set to allow all individuals currently living in Australia to participate, allowing a more representative sample of the general population.

The participants completed a two-part online questionnaire via Qualtrics. In the first part of the survey, participants were presented with a participant information sheet and consent form, where they were required to indicate informed consent by checking a box to continue. They were then required to complete a CAPTCHA to prevent bot participation and enter identifying information. The participants then answered demographic questions and completed measures of attitudes, subjective norms, perceived behavioural control and intention.

One week later, the participants received a link to the second part of the survey. The participants entered their identifying information and indicated their plant-based milk consumption over the previous week. At the end of the survey, the participants were thanked for completing the study. For completing both parts of the study, those recruited from Prolific received GBP 2.00 (equal to or more than minimum wage) and those recruited from the undergraduate participant pool received course credit. Both surveys took approximately 15–20 min in total to complete.

### 2.2. Participants

Five hundred and forty-eight individuals completed the first part of the study, while four hundred and eight participants completed the second part of the study. Of the participants who completed both parts of the study, 122 participants were excluded for missing identifying information to match data from both parts of the study, being a duplicate response or not indicating their behaviour. Two hundred and eighty-six participants were retained in the final sample. A sensitivity analysis conducted using G*Power version 3.1.9.7 [37] indicated that our final sample of 286 participants was sufficiently powered to detect a small to moderate effect size (*f^2^* = 0.06) with α = 0.05, power at 0.80 and 10 predictors. The effect size was also guided by past research indicating that a small effect was sufficiently powered to identify significant results [25]. The participants were aged between 18 and 83 years (*M* = 30.99, *SD* = 10.51). Most resided in Australia (99.7%) and identified as women (55.9%), while 42.7% identified as men and 1.4% identified as non-binary or another gender. See Table 1 for a breakdown of participant demographics. Ethics approval was obtained from the Curtin University Human Research Ethics Committee (HRE2024-0433).

### 2.3. Measures

Items assessing attitude, subjective norms and behavioural beliefs were developed based on the theory of planned behaviour measure development guide by Ajzen [38]. Items were developed for this study due to the lack of existing standardised measures of the theory of planned behaviour and recommendations in the theory to create items specific to the behaviour and context of interest [19,38]. Theory constructs were also expanded to consider environmental, health and taste motives of plant-based milk consumption. Individual items of each theory construct measure assessed each of the three main motives of plant-based milk consumption. Following the guidelines to specify the behaviour and context of interest, we incorporated the environmental, health and taste motives of plant-based milk consumption identified in the broader literature. Therefore, our findings may extend the application of the theory to consider the motives of plant-based milk consumption, which were not previously explored.

#### 2.3.1. Attitude

The participants responded to three items assessing their environmental- (“Plant-based milk is good for the environment”), health- (“Plant-based milk is healthy”) and taste-focused attitudes (“Plant-based milk tastes good to me”) towards plant-based milk. The items were answered on a 7-point Likert scale (1 = strongly disagree to 7 = strongly agree). Higher scores indicated more positive attitudes towards plant-based milks.

#### 2.3.2. Subjective Norms

The participants responded to three items assessing their perceptions of others’ approval of plant-based milk consumption due to environmental (“People whose opinions I value consume plant-based milk to be environmentally conscious”), health (“People whose opinions I value consume plant-based milk because it is good for their health”) and taste motives (“People whose opinions I value consume plant-based milk because it tastes good”). The items were answered on a 7-point Likert scale (1 = strongly disagree to 7 = strongly agree). Higher scores indicated stronger perceptions that others support plant-based milk consumption.

#### 2.3.3. Behavioural Beliefs

The participants answered three items assessing their confidence in positive environmental (“I am confident that consuming plant-based milk will improve environmental outcomes”), health (“I am confident that consuming plant-based milk will improve my health”) and taste outcomes (“I am confident that consuming plant-based milk will taste good”). The items were answered on a 7-point Likert scale (1 = strongly disagree to 7 = strongly agree). Higher scores indicated greater confidence in the positive environmental, health and taste outcomes of plant-based milk consumption.

#### 2.3.4. Intention

The participants answered one item assessing their intention to consume plant-based milk (“I intend to consume plant-based milk instead of dairy milk over the week”) on a 7-point Likert scale (1 = strongly disagree to 7 = strongly agree). Higher scores indicated stronger intention to consume plant-based milk. Only one item was used to assess intention because previous studies that used multiple items to assess intention found high correlations between these items; thus, we reduced the participant burden [39,40]. Although single-item measures are often criticised for their vulnerability to measurement errors, single-item measures of the theory of planned behaviour have been shown to have good content validity, high correlations and similar predictive ability to multi-item theory measures [41]. Single-item measures are acceptable for constructs that are unidimensional, clearly defined and narrow in scope [42].

#### 2.3.5. Plant-Based Milk Consumption

The participants answered one question to indicate their consumption of plant-based milk over the previous week (“Over the past week, how often have you consumed plant-based milk instead of dairy milk?”) in a multiple-choice format (1 = “I have not consumed plant-based milk over the past week” to 4 = “Every day over the past week”). Higher scores indicated greater consumption of plant-based milk.

### 2.4. Data Analysis

The data was analysed using two multiple regression analyses using IBM SPSS Statistics Version 29 [43]. The general form of the multiple regression formula used wasY = β_0_ + β_1_X_1_ + β_2_X_2_ + … + ε

In both models, Y is the outcome variable; β_0_ is the intercept; β_1_, β_2_, … are the unstandardised regression coefficients; X_1_, X_2_, … are the predictor variables; and ε is the error term.

The first multiple regression analysis was used to determine the variance in intention that could be explained by environmental-, health- and taste-focused attitudes; subjective norms; and behavioural beliefs. All predictors were simultaneously entered into the model to assess their unique contributions. In the first multiple regression analysis, the specific regression formula wasIntention = β_0_ + β_1_(environmental attitude) + β_2_(health attitude) + β_3_(taste attitude) + β_4_(environmental subjective norm) + β_5_(health subjective norm) + β_6_(taste subjective norm) + β_7_(environmental behavioural beliefs) + β_8_(health behavioural beliefs) + β_9_(taste behavioural beliefs) + ε

The second multiple regression analysis was used to determine the variance in plant-based milk consumption that could be explained by intention and environmental-, health- and taste-focused behavioural beliefs. All predictors were simultaneously entered into the model to assess their unique contributions. In the second multiple regression analysis, the specific regression formula wasPlant-based milk consumption = β_0_ + β_1_(intention) + β_2_(environmental behavioural beliefs) + β_3_(health behavioural beliefs) + β_4_(taste behavioural beliefs) + ε

## 3. Results

On average, the participants consumed plant-based milk once over the one-week period (*M* = 2.21, *SD* = 10.51). See Table 2 for descriptive statistics and bivariate correlations between variables.

### 3.1. Predicting Intention

Environmental-, health- and taste-focused attitudes; subjective norms; and behavioural beliefs were simultaneously entered into the model. The overall model was significant and accounted for 53.2% of the variance in intention; *R*^2^ = 0.53, *F* (9, 276) = 34.89, *p* < 0.001. This indicated that the model provided a significantly better fit to the data than a model with no predictors and that it could explain a meaningful amount of variation in intention. Taste-focused attitude and health-focused behavioural beliefs were the only significant unique predictors. See Table 3 for the contributions of each variable in the model.

### 3.2. Predicting Plant-Based Milk Consumption

Environmental-, health- and taste-focused intention and behavioural beliefs were simultaneously entered into the model. The overall model was significant and accounted for 60.3% of the variance in plant-based milk consumption; *R*^2^ = 0.60; *F* (4, 281) = 106.58, *p* < 0.001. This indicated that the model provided a significantly better fit to the data than a model with no predictors and that it could explain a meaningful amount of variation in behaviour. Intention was the only significant unique predictor. See Table 4 for the contributions of each variable in the model.

## 4. Discussion

The aim of this study was to explore the factors influencing the intention to consume and consumption of plant-based milk using the theory of planned behaviour (attitudes, subjective norms and behavioural beliefs). The second aim was to determine whether environmental, health or taste perspectives of the theory constructs were most likely to influence intention and consumption regarding plant-based milk. The findings provide partial support for Hypothesis 1, as taste-focused attitude (H1a) and health-focused behavioural beliefs (H1c) were significant predictors of intention, while subjective norms (H1b) was not. Similarly, the findings provide partial support for Hypothesis 2, as intention (H2a) was a significant predictor of plant-based milk consumption, while behavioural beliefs (H2b) was not. However, Hypothesis 3 was not supported, as environmental-focused attitudes (H3a), subjective norms (H3b) and behavioural beliefs (H3c) were not significant predictors of intention to consume plant-based milk.

Taste-focused attitude was a significant predictor of intention, while environmental- and health-focused attitudes were not. It is surprising that environmental- and health-focused attitudes were not significant predictors of intention, as previous research indicated that environmental and health motives were major factors of plant-based milk consumption [5]. Health and environmental motives are typically more prominent among health-conscious and environmentally-conscious consumers [44]. However, the participants in this study only expressed moderately positive attitudes toward the environmental (*M* = 4.92, *SD* = 1.47) and health (*M* = 4.23, *SD* = 1.54) benefits of plant-based milk. For individuals who do not strongly value these benefits, taste may become more influential. This may be because hedonic values often outweigh more conscious motives due to pleasure being associated with food choice [45]. However, the findings support the literature indicating that the acceptance and likeability of taste is still the most important factor influencing animal- or plant-based food choice [14] and that other attributes of dairy alternative products cannot compensate for disagreeableness in taste [17]. Therefore, plant-based milk manufacturers should prioritise promoting the taste of products to target individuals with varying taste preferences to strengthen their intention to consume. Manufacturers and marketers may consider the taste profiles of consumers. For example, those who prefer dairy favour milk-like and sweet tastes but not weak or bland flavours, while those who prefer dairy and tri-blend alternative milks favour coconut-like, creamy flavours and thick consistencies but not cardboard-like, bland tastes or watery consistencies [18].

Subjective norms focusing on the environment, health or taste were not significant predictors of intention. These findings are consistent with research indicating that subjective norms did not influence consumers’ intention to consume plant-based yogurt alternatives [25]. However, the findings are inconsistent with research indicating that social influences affected adolescents’ consumption of alternative milks [46]. It is possible that subjective norms were not influential in the current study as subjective norms tend to be more influential for younger people [46,47], if the behaviour is not socially approved [48] or if behaviours may directly impact others (e.g., safer sex behaviours; [47]). Plant-based milks are often marketed as ethical and socially conscious alternatives to dairy products [49]. But the benefits of environmentally friendly behaviours may not be immediately seen [50]. Therefore, marketers should target more influential drivers of plant-based milk consumption that focus on the beneficial outcomes for the individual.

Health-focused behavioural beliefs were a significant predictor of intention, while environmental- and taste-focused behavioural beliefs were not. These findings support previous research indicating that health is an important consideration for consumers and that consumers think that plant-based milk is healthier than dairy milk [51,52]. Although the findings contradict previous research indicating that individuals consume plant-based diets to reduce their environmental impact, consumers are often more driven by personal health benefits than abstract or global concerns [24,53]. Health outcomes (e.g., improved digestion from avoiding dairy) are often directly observable and perceived to be within personal control. Contrastingly, environmental outcomes (e.g., reducing greenhouse gas emissions) may be influenced by systemic factors and feel less controllable. Therefore, environmentally focused behavioural beliefs may not be significant in this study as people are generally more motivated by outcomes they view as controllable [54]. Similarly, it is possible that individuals perceive that they cannot control for the taste of plant-based milk, as individuals who find plant-based milk acceptable are less impacted by the commonly unfavourable taste attributes of plant-based milks compared to those who prefer full dairy products [18]. Therefore, other outcomes like health benefits, which may be more controllable and personally relevant, may become more salient when deciding to consume plant-based milk. Due to this, marketers should target individuals’ health-focused behavioural beliefs to strengthen their intentions to consume plant-based milk by increasing their confidence to improve their health (e.g., by providing resources to more easily integrate plant-based milks into diets and by emphasising the controllability of health outcomes by consuming plant-based milk).

Although intention was a significant predictor of plant-based milk consumption, environmental-, health- and taste-focused behavioural beliefs were not. These findings are consistent with meta-analyses indicating that intention is the largest predictor of behaviour [47,55]. However, it should be noted that behavioural beliefs, rather than perceived behavioural control, were assessed in the current study. This amendment was due to prior research suggesting that perceived behavioural control may be more important for behaviours with near-future outcomes, rather than distant-future outcomes [28]. Contrastingly, behavioural beliefs may be more important for behaviours with distant-future outcomes [29], such as plant-based milk consumption. But it was proposed in the theory of planned behaviour that perceived behavioural control will predict intention and behaviour [19]. Therefore, future research should consider the impact of environmental-, health- and taste-focused perceived behavioural control to determine whether these control-related beliefs are influential in plant-based milk consumption. However, as the findings indicate that intention is a significant predictor, marketers should prioritise strengthening consumers’ intentions to consume plant-based milk, for example, by targeting taste-focused attitudes and health-focused behavioural beliefs (as the significant predictors of intention) by emphasising the favourable taste aspects of plant-based milks, as well as the potential to control for health outcomes via consuming plant-based milks.

### Strengths and Limitations

The current study provides evidence to support the predictive ability of the theory of planned behaviour to explain factors driving the intention to consume and consumption of plant-based milks. Prior research (e.g., [24,25]) has highlighted the role of the theory of planned behaviour in plant-based dietary decision making or identified the motives via exploratory methods (e.g., [33,34]). However, no studies to date have incorporated the relevant motives within a theoretical framework to provide a comprehensive, theory-driven investigation of plant-based milk consumption. This study provides partial support for the extended theory of planned behaviour in plant-based milk consumption, which was not previously explored. This study also provides insights into the unique contributions of the common motives driving plant-based milk consumption. Therefore, theory-driven and individualised approaches that consider personal dietary motives may be explored to promote the consumption of plant-based milk and other sustainable dietary behaviours.

The findings also offer valuable insights for marketing strategies aimed at promoting the consumption of plant-based milk and similar environmentally conscious foods. With the plant-based food market becoming increasingly competitive, brands that are currently focusing on sustainability messaging may instead consider the findings of this study to focus on consumer groups’ taste profiles [18] or a health-centred approach [56]. This shift could resonate more with consumers who prioritise the taste aspects of plant-based milks or those who prioritise controlling for personal health benefits, as these motives were identified as significant predictors of the intention to consume plant-based milk.

However, the self-report method to record participants’ plant-based milk consumption is a limitation to the current study, as participants commonly report inaccuracies when reporting dietary behaviours [57]. Future studies could address this by employing objective measures of behaviour (e.g., via observational studies) or through the use of the Timeline Follow-back measure [58], which implements a calendar format to improve memory recall of past behaviours [59].

Additionally, while this study explored health-focused perspectives of the theory of planned behaviour constructs, the role of lactose intolerance, which could significantly influence plant-based milk consumption, was not controlled for. Including this variable in future studies could provide more nuanced insights into the motivations for choosing plant-based milk over dairy alternatives [12]. Further, future studies could more systematically capture the nuances of health-related food choices by applying standardised measures to identify the specific health benefits consumers associate with plant-based milk.

The use of convenience sampling in this study limits the generalisability of the findings to the broader population. While participants were recruited through Prolific (an online recruitment platform that provides access to a diverse pool), they were also recruited from an undergraduate participant pool. Therefore, the sample may primarily reflect the perspectives of individuals from Western, educated, industrialised, rich and democratic (WEIRD) societies, which are commonly represented in university populations [60]. However, research has suggested that including crowdsourced samples (e.g., via online platforms) can offer greater diversity in age, ethnicity, race, education and gender [61]. Future studies could replicate this research with more representative samples to better capture the views of the general population.

Finally, behavioural beliefs were assessed instead of perceived behavioural control in this study. Research has highlighted the potential value in assessing behavioural beliefs for behaviours with distant-future outcomes [29], such as plant-based milk consumption. However, this deviates from the original theory of planned behaviour, where perceived behavioural control was instead proposed to influence intention and behaviour [19]. Therefore, future research should also determine whether perceived behavioural control and associated control beliefs may be influential in plant-based milk consumption. Nevertheless, by including behavioural beliefs in this study, it was possible to highlight the role of health-related behavioural beliefs in individuals’ intentions to consume plant-based milk.

## 5. Conclusions

The aim of this study was to explore the factors influencing the intention to consume and consumption of plant-based milk using the theory of planned behaviour (attitudes, subjective norms and behavioural beliefs). Additionally, the second aim was to determine whether environmental, health or taste perspectives of the theory constructs were most likely to influence the intention to consume and consumption of plant-based milks. Taste-focused attitudes and health-focused behavioural beliefs were identified as key factors of intention. Intention was identified as the sole predictor of plant-based milk consumption. These findings indicate that future behaviour change interventions should focus on strengthening consumer intentions to encourage plant-based milk consumption by emphasising the taste value and by instilling individuals’ confidence to attain health benefits. The results could direct more effective marketing strategies, public health initiatives and consumer education campaigns that place greater emphasis on the taste and tangible health benefits of plant-based products.

## Figures and Tables

**Figure 1 foods-14-01970-f001:**
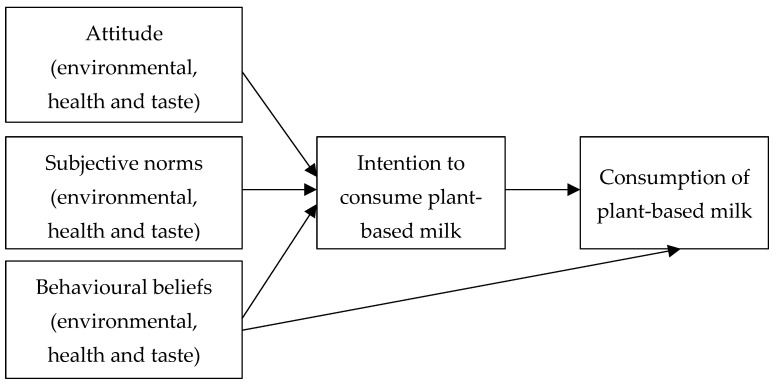
Extended theory of planned behaviour including environmental, health and taste frames. Note: Adapted from Ajzen [19].

**Table 1 foods-14-01970-t001:** Participant demographics (*N* = 286).

	*N*	%
Age (years)		
18–39	223	78.0
40–64	59	20.6
65 and over	2	0.7
Did not indicate	2	0.7
Gender		
Men	122	42.7
Women	160	55.9
Non-binary	4	1.4
Australian residence		
Yes	285	99.7
No	1	0.3

**Table 2 foods-14-01970-t002:** Descriptive statistics and bivariate correlations between constructs and plant-based milk consumption (*N* = 286).

	*M (SD)*	1	2	3	4	5	6	7	8	9	10	11	12	13
1. Gender	-	-	−0.09	−0.03	0.08	0.15 *	0.07	0.12 **	0.22 **	−0.01	0.09	0.09	0.16 **	0.14 *
2. Age	30.99 (10.51)		-	−0.03	−0.01	0.05	−0.05	−0.08	−0.01	−0.01	−0.03	0.04	0.13 *	0.15 *
3. Attitude—E	4.92 (1.47)			-	0.55 **	0.38 **	0.47 **	0.24 **	0.26 **	0.80 **	0.48 **	0.41 **	0.38 **	0.31 **
4. Attitude—H	4.23 (1.54)				-	0.48 **	0.32 *	0.40 **	0.26 **	0.53 **	0.76 **	0.52 **	0.48 **	0.38 **
5. Attitude—T	4.06 (1.97)					-	0.28 **	0.21 **	0.41 **	0.41 **	0.48 **	0.90 **	0.69 **	0.57 **
6. SNs—E	4.13 (1.65)						-	0.56 **	0.53 **	0.49 **	0.27 **	0.28 **	0.26 **	0.22 **
7. SNs—H	4.38 (1.63)							-	0.55 **	0.27 **	0.46 **	0.22 **	0.23 **	0.21 **
8. SNs—T	3.91 (1.54)								-	0.33 **	0.29 **	0.41 **	0.33 **	0.26 **
9. BBs—E	4.71 (1.58)									-	0.53 **	0.47 **	0.39 **	0.29 **
10. BBs—H	4.21 (1.53)										-	0.52 **	0.52 **	0.41 **
11. BBs—T	4.01 (1.99)											-	0.68 **	0.53 **
12. Intention	4.15 (2.25)												-	0.78 **
13. Behaviour	2.21 (1.17)													-

Note. E = environment, H = health, T = taste, SNs = subjective norms, BBs = behavioural beliefs. * *p* < 0.05, ** *p* < 0.01.

**Table 3 foods-14-01970-t003:** Individual contributions of variables in predicting intention to consume plant-based milk (*N* = 286).

	B [95% CI]	SE	β	sr^2^
Attitude—E	0.10 [−0.12, 0.32]	0.11	0.07	0.00
Attitude—H	0.02 [−0.18, 0.22]	0.10	0.01	0.00
Attitude—T	0.48 [0.26, 0.70] **	0.11	0.42	0.03
SNs—E	0.04 [−0.12, 0.20]	0.08	0.03	0.00
SNs—H	−0.03 [−0.19, 0.14]	0.08	−0.02	−0.00
SNs—T	0.02 [−0.13, 0.18]	0.08	0.02	0.00
BBs—E	−0.07 [−0.28, 0.14]	0.11	−0.05	−0.00
BBs—H	0.31 [0.11, 0.52] *	0.10	0.21	0.01
BBs—T	0.19 [−0.04, 0.41]	0.11	0.16	0.00

Note. E = environment, H = health, T = taste, SNs = subjective norms, BBs = behavioural beliefs. * *p* < 0.05, ** *p* < 0.01.

**Table 4 foods-14-01970-t004:** Individual contributions of variables in predicting plant-based milk consumption (*N* = 286).

	B [95% CI]	SE	β	sr^2^
Intention	0.40 [0.34 0.45] **	0.03	0.77	0.30
BBs—E	−0.02 [−0.09, 0.04]	0.03	−0.03	−0.00
BBs—H	0.02 [−0.05, 0.10]	0.04	0.03	0.00
BBs—T	0.00 [−0.06, 0.07]	0.03	0.01	0.00

Note. E = environment, H = health, T = taste, SNs = subjective norms, BBs = behavioural beliefs. ** *p* < 0.01.

## Data Availability

The original contributions presented in the study are included in the article. Further inquiries can be directed to the corresponding author.

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
