# Peer review of "Health, Environment or Taste? Using the Theory of Planned Behaviour to Predict Plant-Based Milk Consumption"

_foods, 2025, doi:10.3390/foods14111970_

Round 1

Reviewer 1 Report

Comments and Suggestions for Authors

Thank you very much for submitting your manuscript to this journal. The manuscript employs Ajzen's theory of planned behavior to examine the consumption intention of Plant-Based Milk Consumption. The authors argue that it is desirable not to consume cow's milk because intensive livestock farming has adverse effects on the environment. 

Although the manuscript may be of some interest to the community, it has some limitations that need to be addressed:

  1. This is the most important limitation. The authors do not argue why this manuscript should be published. What scientific interest does it have for the community? How is it different from all previous research? What is the theoretical contribution?
  2. This other limitation cannot be solved simply: It lies in the way perceived behavioural control (PBC) is operationalized. In your study, PBC is measured as participants’ confidence in achieving positive outcomes (environmental, health, or taste benefits) from consuming plant-based milk, rather than their perceived ease or difficulty of actually performing the behaviour itself. This approach diverges from the original TPB framework, where PBC is intended to reflect the perceived controllability over the behaviour (i.e., the act of consuming plant-based milk), not the outcomes that might result from it. This distinction is important because outcome-based and behaviour-specific perceptions of control are not equivalent, and this may affect both the interpretation of your findings and the theoretical validity of your model. I strongly recommend revising the PBC items to align with the TPB’s conceptualization or, at minimum, providing a robust justification and discussion of the implications of this measurement choice.
  3. Instead of regression, the authors should use structural equation analysis to understand the direct and indirect influence of all variables.
  4. The intention to consume plant-based milk is assessed with a single item. While you reference previous studies to justify this, relying on a single-item measure for such a central construct can limit both the reliability and validity of your results. Whenever possible, multi-item scales are preferable, especially for psychological constructs. If using a single item is necessary, please provide stronger justification and acknowledge the limitation more explicitly. en mi opinión, parece que usted usó un solo ítem porque ya tenía en mente que no iba a emplear SEM.
  5. The results section reveals that only taste-focused attitudes and health-focused PBC significantly predict intention, while environmental motives and subjective norms are not significant predictors. This finding is somewhat surprising, given the frequent association between plant-based milk and environmental sustainability in the literature. I encourage you to elaborate further on possible explanations for this discrepancy, such as cultural factors, sample characteristics, or the relative importance of hedonic versus abstract motives in food choice.
  6. Regarding the sample, while the power analysis is appreciated, the use of a convenience sample from Prolific and a university participant pool may limit the generalizability of the findings. I recommend discussing this limitation more thoroughly and considering its implications for the external validity of your conclusions. en realidad es una práctica muy extendida porque no los recursos son limitados, pero al menos mencione eso en las limitaciones.

On a positive note, the manuscript is generally well written, with clear and appropriate academic English. The structure is logical, and the argumentation is coherent. Some sentences, particularly in the discussion section, could be made more concise for clarity, but overall the language is suitable for publication.

I have reason to reject the manuscript, although I will consider major changes. The authors may want to address the review if they have the questionnaire data for it, but the authors should consider it a risky review.

Reviewer 2 Report

Comments and Suggestions for Authors

The paper is interesting but it needs to be strengthened in some parts, primary the methodological one. In detail my comments:

  • The introduction section is interesting and explain the context in which the paper is put, however, Authors should improve it showing better the gap present in literature about previous studies on this issue, in order to better underline the value of the research.
  • In hypothesis development Authors say: “Attitude, subjective norm, and perceived behavioural control will explain significant 115 variance in intention to consume plant-based milk.” But attitude, subjective norm, and perceived behavioural control are 3 different variables and in my opinion each one should be separated in a single hypothesis or divede H1 in H1a, H1b, H1c. The same for H2 e H3, a single Hypothesis should contain a single variable.
  • In the methodology section it woud be relevant to explain what “Prolific” is and to better clarify how sampling has been done, otherwise how can we understand is the sample reached could be generalizable of an entire population, a group of it or if these results obtained are not generalizable? Authors should be more rigorous in describing sampling procedures and underline its limitations if needed.
  • Always in the methodology section there is a small description of socio demographic features of respondents but it should be relevant to insert a table defining in detail ages and profile of respondents in order to understand the characteristics of the sample analysed.
  • Authors said “Items were developed based on recommendations by Ajzen [33] and Francis, Johnston 160 [34] to construct a theory of planned behaviour questionnaire and framed to consider 161 environmental, health and taste motives of plant-based milk consumption. Individual 162 items of each theory construct measure assessed each of the three main motives of plant-163 based milk consumption.”, ok for usingtheory of planned behaviour to construct items, but did these items (motivations to consume plant-based milk were drawn from literature or how did they were constructed? It is necessary that Authors develop a table with items and reference literature of each one.
  • Authors say that “Data was analysed using two multiple regression analyses using IBM SPSS Statistics 202 Version 29”, however they do not show the regression formula. It would be really relevant to show and explain them in the methodology section.
  • Table 1, all acronyms such as Attitude – E…, SN – E…., PBC – E…. are not explained. It is of big relevance that what is written in tables is fully understandable, so please find a way to explain everything.
  • In the discussion section clear reference and discussion to each hypothesis is missing, please integrate the discussion by referring to hypothesis in order to better understand if they are accepted or rejected.
  • The paper misses a section describing the main implications (theoretical and practical) deriving from the results obtained.

Round 2

Reviewer 2 Report

Comments and Suggestions for Authors

Authors have done a good work in reviewing the paper, but I still continue to think that the 2 regression formula, inserting in each the specific variables should be shown in the methodology section together with the elements that explain the good fit or not of the regression model.

The other parts are well improved, thank you.
